# The Effects of Rumen-Protected Choline and Rumen-Protected Nicotinamide on Liver Transcriptomics in Periparturient Dairy Cows

**DOI:** 10.3390/metabo13050594

**Published:** 2023-04-26

**Authors:** Xue’er Du, Zhijie Cui, Rui Zhang, Keliang Zhao, Lamei Wang, Junhu Yao, Shimin Liu, Chuanjiang Cai, Yangchun Cao

**Affiliations:** 1College of Animal Science and Technology, Northwest A&F University, Xianyang 712100, China; 2UWA Institute of Agriculture, The University of Western Australia, Crawley, WA 6009, Australia; shimin.liu@uwa.edu.au

**Keywords:** perinatal cows, rumen-protected choline, rumen-protected nicotinamide, transcriptomics, metabolic syndrome

## Abstract

To investigate the effects of rumen-protected choline (RPC) and rumen-protected nicotinamide (RPM) on liver metabolic function based on transcriptome in periparturient dairy cows, 10 healthy Holstein dairy cows with similar parity were allocated to RPC and RPM groups (*n* = 5). The cows were fed experimental diets between 14 days before and 21 days after parturition. The RPC diet contained 60 g RPC per day, and the RPM diet contained 18.7 g RPM per day. Liver biopsies were taken 21 days after calving for the transcriptome analysis. A model of fat deposition hepatocytes was constructed using the LO2 cell line with the addition of NEFA (1.6 mmol/L), and the expression level of genes closely related to liver metabolism was validated and divided into a CHO group (75 μmol/L) and a NAM group (2 mmol/L). The results showed that the expression of a total of 11,023 genes was detected and clustered obviously between the RPC and RPM groups. These genes were assigned to 852 Gene Ontology terms, the majority of which were associated with biological process and molecular function. A total of 1123 differentially expressed genes (DEGs), 640 up-regulated and 483 down-regulated, were identified between the RPC and RPM groups. These DEGs were mainly correlated with fat metabolism, oxidative stress and some inflammatory pathways. In addition, compared with the NAM group, the gene expression level of *FGF21*, *CYP26A1*, *SLC13A5*, *SLCO1B3*, *FBP2*, *MARS1* and *CDH11* in the CHO group increased significantly (*p* < 0.05). We proposed that that RPC could play a prominent role in the liver metabolism of periparturient dairy cows by regulating metabolic processes such as fatty acid synthesis and metabolism and glucose metabolism; yet, RPM was more involved in biological processes such as the TCA cycle, ATP generation and inflammatory signaling.

## 1. Introduction

The peripartum period, lasting from 3 weeks before and 3 weeks after parturition, induces remarkable physiological and metabolic adaptations in mammals which are crucial for a good reproductive performance and to ensure the suitable development of the fetus and to provide adequate substrates that are needed in utero and following birth [1]. Simultaneously, cows may face immune system dysfunction and increased inflammatory status, which are important factors limiting their ability to achieve optimal production performance and immune metabolic status and may have a negative impact on the successful transition of offspring from intrauterine to extrauterine [2]. Perinatal cows are subject to the imbalance between energy demand and dietary energy intake, resulting in stimulation of fat mobilization [3] and an elevated level of non-esterified fatty acids (NEFA) in the body. When the elevated NEFA excesses the clearance limit of the body, there will be excessive accumulation of fat and its metabolites in the liver, such as ketone bodies, leading to fatty liver and ketosis [4]. In addition, the excessive NEFA will also lead to oxidative stress [5], rumen acidosis, protein metabolism imbalance and other metabolic syndromes [6]. These metabolic disorders can increase the incidence of other diseases, such as placental retention and endometritis, which will impair the immune system of cows, reduce the milk yield and reproductive efficiency of cows, and cause serious economic losses [7,8,9]. To counteract these negative effects, rumen-protected products are frequently added to the diet of dairy cows during the periparturient period to maintain the physical condition of dairy cows, enhance productivity performance and reduce economic losses.

Choline (2-hydroxyethyl trimethyl ammonium salt) is an essential nutrient for humans and animals which needs to be obtained from the diet in addition to endogenous synthesis. Choline plays a significant role in cell maintenance and growth at various stages of life, including neurotransmission [10], membrane synthesis, lipid transport and one-carbon metabolism [11]. In the liver, choline can be used to synthesize phosphatidylcholine through the cytidine diphosphate–choline pathway and can also be oxidized to betaine, which, through the phosphatidylethanolamine N-methyl transfer pathway, is used to synthesize very-low-density lipoprotein (VLDL) to participate in lipid transport. It has been found that the imbalance between the proportion of phosphatidylcholine and phosphatidylethanolamine in the liver will affect the integrity of the cell membrane, leading to cell damage and inflammatory reactions such as steatohepatitis [12]. In recent years, accumulating evidence suggests that choline can regulate the metabolism and transport of glucose and lipid, as well as the occurrence of oxidative stress in the body by regulating the expression of related genes and activating signal pathways such as AMPK [13,14,15]. Moreover, a large number of animal experiments have shown that adding choline to the diet of periparturient dairy cows could increase the level of VLDL [16,17], regulate fat metabolism [18], relieve metabolic diseases such as fatty liver and ketosis [19,20], increase milk yield [21,22], improve milk quality [23,24], maintain an animal’s physical condition [25,26], boost immune system [27,28], and also promote the growth performance of offspring [29].

Nicotinamide is a member of water-soluble B vitamins, which can be served as a precursor to synthesize nicotinamide adenine dinucleotide (NAD) and nicotinamide adenine dinucleotide phosphate [30,31]. In the liver, nicotinamide can be converted and synthesized from nicotinic acid. In addition, nicotinamide can also be produced through the hydrolysis of nicotinamide adenine dinucleotide (NADH) and NAD+ by coenzyme β—Nicotinamide adenine dinucleotide (NAD+). These metabolites can be directly used for ATP generation, DNA synthesis and repair, as well as other metabolic pathways and cell signal transductions. Moreover, nicotinamide also plays an important role in the regulation of lipid and energy metabolism, oxidative stress, mitochondrial dysfunction and other processes [32,33]. Plenty of studies indicate that the addition of nicotinamide and niacin can reduce the levels of total cholesterol, triglyceride, high-density lipoprotein (HDL) cholesterol, LDL cholesterol and VLDL cholesterol as well as improve dyslipidemia, alleviate adipose tissue inflammation [34,35,36], and increase milk yield and the milk protein content of lactating dairy cows [37,38].

However, the difference between choline and nicotinamide in liver metabolism and its mechanism in periparturient dairy cows is still unclear. Therefore, in this study, we added rumen-protected choline (RPC) and rumen-protected nicotinamide (RPM) to the diet of periparturient dairy cows. We then performed a liver biopsy and explored the effects of choline and nicotinamide on the liver metabolism of periparturient dairy cows and the differences in their mechanisms based on transcriptome.

## 2. Materials and Methods

### 2.1. Animals and Sample Acquisition

Ten healthy Chinese Holstein pregnant cows with similar parity were selected and randomly divided into 2 groups: the rumen-protected choline group (60 g/d RPC, active ingredient 15 g/d, purity 65%, the rumen bypass rate 80.3%) and the rumen-protected nicotinamide group (18.7 g/d RPM, active ingredient 9.6 g/d, purity 60%, the rumen bypass rate 85.6%). The experimental period lasted for 35 days, from 14 days before parturition to 21 days after calving. During the monitoring period, the cows in health status and 1–2 parity delivered healthy, viable full-term calves.

The feeding and management of cattle were carried out in accordance with the pasture management system. During the experimental period, two diets were prepared, namely, the pre-partum diet and the post-partum diet. The diets were prepared in the form of total mixed rations (TMR) and offered twice a day, at 07:00 and 16:00, respectively. All experimental animals had free access to food and water. Liver biopsy was performed on the 21st day after delivery (collected at 13:00 each time) by using a biopsy needle, and the liver samples collected were quickly stored at −80 °C. All the experimental procedures in the present study were conducted according to the Animal Protection Law based on the Guide for the Care and Use of Laboratory Animals approved by the Ethics Committee of Northwest A&F University.

### 2.2. Culture of Human Hepatocyte Line LO2

LO2 cells were cultured in Roswell Park Memorial Institute (RPMI) 1640 medium (HyClone, Logan, UT, USA) containing 10% fetal bovine serum (Gibco BRL, Grand Island, NY, USA), and the culture was performed at 37 °C under 5% CO2. To establish a cell model of fatty liver in periparturient dairy cows, LO2 cells were plated into 6-well culture plates to which was added 1.6 mmol/L NEFA subject to two treatments: CHO and NAM. The CHO group contained 75 μmol/L choline in choline-free RPMI 1640 medium, and the NAM group contained 2 mmol/L nicotinamide in nicotinamide-free medium. Before treatment, cells were starved using choline-free RPMI 1640 medium (Gibco BRL, Grand Island, NY, USA) and nicotinamide-free medium (Coolaber, Beijing, China), respectively, without adding fetal bovine serum for 6 h. After starving, LO2 cells with experimental treatment were incubated for 24 h. NEFA was formulated from palmitic acid (3.19 mmol), palmitoleic acid (0.53 mmol), stearic acid (1.44 mmol), linoleic acid (0.49 mmol) and oleic acid (4.35 mmol). All reagents without instructions were purchased from Sigma-Aldrich.

### 2.3. RNA Isolation and Quantitative Real-Time Polymerase Chain Reaction

Total RNA in cells was isolated with Trizol (Invitrogen). An amount of 1–2 μg of total RNA was treated with DNAse (Invitrogen) and reverse transcribed into cDNA using M-MLV Reverse Transcriptase (Accubate Biology, Changsha, China). Quantitative Real-Time PCR (RT-PCR) was performed using SYBR^®^ Select Master Mix (Accubate Biology, Changsha, China) and the Roche LightCycler^®^ 480 Real-Time PCR System. The 2^−ΔΔCt^ method was used to determine the relative quantitative gene expression levels, normalized by β-actin (Table 1 and Table 2) [39].

### 2.4. RNA Isolation and cDNA Library Construction

Total RNA in liver biopsies was extracted, and the RNA concentration and purity were determined using a Nanodrop2000 (Thermo Fisher Scientific, US). Agarose gel electrophoresis was used to detect the integrity of RNA. An Agilent 2100 bioanalyzer was used to measure the RNA value. The enriched mRNA was segmented then reverse synthesized to cDNA, which completed the construction of the library to be sequenced. The following conditions must be met for the creation of a single library to be sequenced: the total amount of RNA ≤ 1 µg, the concentration ≥ 35 ng/μL, OD260/280 ≥ 1.8, OD260/230 ≥ 1.0. After passing the quality inspection, the library was sequenced on the IlluminaNovaseq6000 platform, and the database was established by Shanghai Meiji Biomedical Technology Co.

### 2.5. Analysis of Differential Expression of Genes

The software DESeq2 based on negative binomial distribution was used to analyze the differential expression of genes (DEGs). According to *p*-value and |log_2_ FC| as screening conditions, log_2_ FC > 0 indicates gene expression is up-regulated and log_2_FC < 0 indicates gene expression is down-regulated.

### 2.6. Analysis of DEG Enrichment

The software Goatools was used for GO (Gene Ontology) enrichment analysis of genes/transcripts, and R script was used for KEGG (Kyoto encyclopedia of genes and genomes) pathway enrichment analysis of genes/transcripts. Fisher’s Exact Test and the Benjamin–Hochberg method was used to correct the *p*-value. Significance of GO enrichment and KEGG enrichment was considered as a *p*-value < 0.05.

### 2.7. Statistical Analysis

MetaboAnalyst 5.0 (Wishart Research Group, Edmonton, AB, Canada) software was used to perform orthogonal partial least squares-discriminant analysis (OPLS-DA) on the samples from the RPC group and RPM group. A volcano plot was used to analyze DEGs and their changes in each group, and a cluster heatmap was used to analyze the gene expression patterns in each sample. Statistical significance was determined with IBM SPSS Statistics 26.0. Data were represented as an average ± SD and treated with normal distribution test. Student’s *t*-test was used for two-group comparisons. ** indicates *p* < 0.01, *** indicates *p* < 0.001.

## 3. Results

### 3.1. Sequencing Data Quality Control

After sequencing data generation, the original sequencing data for each sample was subjected to a quality assessment such as the base group’s error rate and content distribution, and quality control was carried out to obtain high-quality control data (clean data). The Q20 and Q30 quality values were 97.51% and 93.02%, respectively. The GC content ratio was within an acceptable range, which indicated that the sequencing results were reliable and could be used for subsequent results analysis (Appendix A).

### 3.2. Analysis of DEGs

Transcripts with an absolute value of |log_2_ FC| ≥ 1 and adjusted *p*-value < 0.05 were defined as significant DEGs. A total of 11,023 genes were expressed in all 10 samples, while 828 genes were specifically expressed in the RPC group and 481 were specifically expressed in the RPM group (Appendix A). From the comparisons between these two groups, 1123 DEGs were identified, 640 up-regulated and 483 down-regulated (Figure 1; Appendix A). The cluster heatmap analysis was performed on the identified DEGs, as shown in Figure 2, which shows that the samples from the RPC and RPM groups had obvious clustering. OPLS-DA analysis showed that the samples from the RPC and RPM groups were significantly separated (Figure 3).

Table 3 listed the top 30 DEGs, including *FARS2*, *NOTUM*, *CYP2C19*, *KCNN2*, *CYP26A1*, *SLCO1B3*, *HRG*, *CES1*, *CD1D*, *FBP2*, *SEC14L3*, *PLEK*, *TKT*, *BIRC5*, *FGF21*, *FUT1*, *CDH11*, *DTX1*, *ATP6V1C2*, *SLC13A5*, *LRRC73*, *HOPX*, *MICAL2*, *GPC3*, *ADCY2*, *GLI1*, *MARS1*, *INHBE*, *CARS1*, *CITED4*, etc. The genes with up-regulated expression levels included *FARS2*, *NOTUM*, *CYP2C19*, *KCNN2*, *CYP26A1*, *SLCO1B3*, *HRG*, *CES1*, *CD1D*, *FBP2*, *SEC14L3*, *PLEK*, *TKT*, *BIRC5*. The genes with down-regulated expression levels included *FGF21*, *FUT1*, *CDH11*, *DTX1*, *ATP6V1C2*, *SLC13A5*, *LRRC73*, *HOPX*, *MICAL2*, *GPC3*, *ADCY2*, *GLI1*, *MARS1*, *INHBE*, *CARS1*, and *CITED4*.

### 3.3. Analysis of Differential Gene Enrichment

The GO analysis was performed to detect the function of the identified DEGs. The 268 DEGs were enriched in 44 GO terms, including 20 terms in biological process (BP), 13 terms in cellular component (CC), and 10 terms in molecular function (MF) (Figure 4). Table 4 shows the top 20 items under GO enrichment which are mainly involved with terms relating to fat anabolic processes, including lipid metabolic process, small molecule metabolic process, cellular hormone metabolic process, xenobiotic metabolic process, extracellular space, monooxygenase activity, organic acid metabolic process, oxoacid metabolic process, hormone metabolic process, carboxylic acid metabolic process, monocarboxylic acid metabolic process, oxidoreductase activity, tetrapyrrole binding, chemokine activity, steroid metabolic process, detoxification, fatty acid omega-hydroxylase activity, steroid hydroxylase activity, etc.

Pathway-based analysis was conducted via KEGG to ulteriorly investigate the biological functions of these DEGs. A total of 27 pathways significantly enriched in two groups were identified, which are mainly related to fat metabolism and oxidative stress (Table 5), including retinol metabolism, metabolism of xenobiotics by cytochrome P450, steroid hormone biosynthesis, pyrimidine metabolism, bile secretion, drug metabolism - cytochrome P450, viral protein interaction with cytokine and cytokine receptor, linoleic acid metabolism, arachidonic acid metabolism, chemokine signaling pathway, aminoacyl-tRNA biosynthesis, longevity regulating pathway - worm, inflammatory mediator regulation of TRP channels, pentose phosphate pathway, chemical carcinogenesis - reactive oxygen species, purine metabolism, glutathione metabolism, GnRH secretion, cortisol synthesis and secretion, ovarian steroidogenesis, ascorbate and aldarate metabolism, complement and coagulation cascades, etc. (Figure 5).

### 3.4. Effects of Choline and Nicotinamide on Gene Expression Related to Liver Metabolism

Based on the above analysis results, in order to further verify the effect of choline and nicotinamide on liver metabolic function, the mRNA expression of some DEGs (*FGF21*, *CYP26A1*, *SLC13A5*, *SLCO1B3*, *FBP2*, *MARS1*, *CDH11*) associated with liver metabolic function was tested. As shown in Figure 6, compared with the NAM group, the gene expression of *FGF21*, *CYP26A1*, *SLC13A5*, *SLCO1B3*, *FBP2*, *MARS1*, *CDH11* in the CHO group increased significantly (*p* < 0.05).

## 4. Discussion

The severe negative energy balance in periparturient dairy cows stimulates a large amount of body fat mobilized to support the energy demand of animals, resulting in the disorder of fat metabolism, particularly in the liver. Numerous studies have found that dietary choline and dietary nicotinamide supplementation can effectively alleviate the occurrence of metabolic diseases such as fatty liver, oxidative stress and inflammation [40,41,42]. This study compared the transcriptomic differences in liver metabolism in periparturient dairy cows with the addition of RPC and RPM, identified differential genes and pathways, and discussed the differences between choline and nicotinamide on liver metabolism at the transcriptome level.

In the present study, 640 genes up-regulated by the addition of RPC or RPM and 483 genes down-regulated by the addition of RPC or RPM were identified; the top 20 GO items were mainly involved in fat anabolic processes. The top two DEGs were phenylalanyl-tRNA synthetase 2 and fibroblast growth factor 21. Fibroblast growth factor 21 (*FGF21*), a member of the fibroblast 19 subfamily, is synthesized by the liver and regulated by peroxisome proliferators-activated receptors-α, carbohydrate response element binding protein and other reaction elements. It can act directly on liver cells as a paracrine or autocrine hormone and can also enter the systemic circulation and act on other tissues and organs to regulate glucose, lipid and energy metabolism [43,44]. Some researchers fed *FGF21* knockout mice with a ketogenic diet and found that fat levels in the plasma of the mice were increased, obvious steatosis occurred, and the regulatory function of ketogenesis and sugar metabolism was impaired [45,46]. The chronic accumulation of triglycerides in the liver improves the β-oxidation rate of fatty acids in liver cells. When the energy supply from fatty acid oxidation exceeds the energy demand, it leads to the formation of reactive oxygen species, which, in turn, cause oxidative stress, thus inducing an increase in the synthesis of FGF21 [47,48]. It is worth noting that FGF21 can respond to low protein levels in the body to increase its own circulation and increase energy consumption to cope with the increase in calorie intake [44,49]. The results of the gene expression levels of *FGF21* between two treatment groups showed that choline probably produces an effect on liver fat metabolism and reduces body energy consumption by up-regulating the level of *FGF21*. However, the reason for the reverse change of *FGF21* expression under the influence of choline and nicotinamide needs to be further explored.

The top KEGG pathway of the DEGs in response to the RPC and RPM addition was for retinol metabolism in this study. Vitamin A can be stored in hepatic stellate cells in the form of retinol and provide vitamin A for various tissues of the body. All-trans-retinoic acid is an important active metabolite of vitamin A, and excessive retinoic acid levels in the body will cause changes in physiological functions of the body [50,51]. Retinoic acid hydroxylase CYP26A1 is very important for decomposing exogenous retinoic acid [52]. The results of the cell experiment show that the gene level of *Cyp26a1* in the CHO group was significantly higher than that in the NAM group, indicating that choline can regulate retinoic acid hydroxylase to regulate liver retinol metabolism and maintain its level balance.

Subsequently, the genes closely related to liver metabolism among the DEGs to clarify the differences in the regulation of liver metabolism by choline and nicotinamide were further discussed. The solute carrier family 13 member 5 (SLC13A5) used to mediate citrate transport is mainly expressed in the liver of mammals. As a key intermediate of the tricarboxylic acid (TCA) cycle, citrate plays a key role in the cell metabolism of carbohydrates and fatty acids, as well as energy generation in mitochondria, and can also be served for the biosynthesis of triglycerides, fatty acids, cholesterol and low-density lipoprotein [53,54]. Brachs et al. (2016) found that in the Slc13a5 knockout of mice fed with a high-fat diet, tolerance to insulin resistance and fat deposition induced by the high-fat diet was increased [55]. The expression of Slc13a5 was also increased in the model of type-2-diabetes rats [56]. In addition, some researchers found that the level of intracellular citrate and phospholipid was significantly reduced after silencing the Slc13a5 in HepG2 and Huh7 cells [57]. Solid carrier organic animion transporter family member 1B3 (SLCO1B3, also known as OATP1B3) is a member of the solute transporter superfamily, which is mainly expressed in the basolateral membrane of hepatocytes. It can transport a variety of endogenous and xenobiotic compounds (such as prostaglandins, steroid hormone conjugates, bilirubin, bile acid, and thyroid hormone) to liver cells for metabolism [58,59]. It has been found that the new splice of *OATP1B3*, cancer-type *OATP1B3*, can up-regulate the expression of carnitine palmitoyl transfer and NADH: ubiquinone oxidoreductase subunit A2 through interaction with insulin-like growth factor 2 mRNA-binding protein 2, thereby promoting fatty acid β-oxidative and mitochondrial oxidative phosphorylation activities and increasing ATP production. This results in the formation of plate-like pseudopodia and migration and the invasion of high-grade serous ovarian cancer cells [60]. Sec14-like proteins belong to typical class III phosphatidylinositol transfer proteins, which can transfer phosphatidylinositol and PC between different biological membranes, exchange phosphatidylinositol to PC, and vice versa, so as to maintain membrane–lipid balance [61,62]. This study found that the expression levels of *SLC13A5* and *SLCO1B3* in the CHO group were significantly higher than that in the NAM group, showing that choline may regulate energy metabolism and material transport of the liver by up-regulation of *SLCO1B3*, thereby improving fat metabolite accumulation and energy imbalance. Nicotinamide may reduce the accumulation of citrate by promoting the TCA cycle, allowing the metabolites of citrate, acetyl coenzyme A and oxaloacetate to flow towards ATP synthesis rather than fatty acid synthesis, resulting in the lower gene expression of *SLC13A5*.

Analysis of DEGs revealed a significant increase in the expression levels of FBP and transaminase in the RPC group. Fructose diphosphatase (FBP) can not only regulate the glucose/glycogen synthesis of carbohydrate precursors but can also interact with proteins (such as ATP synthase, HIF1-α and NF-κB) to influence cell-cycle-dependent events, mitochondrial biogenesis and membrane polarization, expression of glycolytic enzymes, induction of synaptic plasticity, and even cancer progression [63]. Transketolase, a key enzyme of the pentose phosphate pathway that is ubiquitous in all organisms, provides a special connection between glycolysis and the non-oxidation period of the pentose phosphate pathway [64]. The study of LO2 cells suggests that the expression level of FBP significantly increased under the regulation of choline, indicating that choline may regulate the process of liver glucose metabolism through FBP and transketolase.

Methyl tRNA synthetase 1 (MARS1) and glycyl tRNA synthetase 1 (GARS1) belong to aminoacyl tRNA synthetases (ARSs). Aminoacyl tRNA synthetases, one of the potential markers of pneumonia, are generally used for protein synthesis and can interact with proteins in the signal pathway of the mammalian target of rapamycin 1, cyclin-dependent kinases 4 and the vascular endothelial growth factor receptor [65,66,67]. Several studies have found that inhibitors of aminoacyl tRNA synthetases have been used in the treatment of many diseases [68,69,70]. Fibroblasts will lose the function of matrix remodeling under pathological conditions, leading to tissue destruction and fibrosis [71,72]. Cadherin (cadherin-11, *CDH11*) is a mesenchymal cadherin that can be expressed in fibroblasts of various tissues and is also a marker and functional regulator of fibroblasts and can mediate homotypic cell adhesion that is important in histomorphogenesis and structure [73,74]. In *CDH11*-knocked diet-induced obese mice, inflammation was reduced and blood glucose levels were controlled [75]. The result of cell assays found that, compared with the CHO group, the levels of formyl tRNA synthetase 1 and *CDH11* genes in the NAM group decreased significantly. Thus, it can be concluded that nicotinamide may affect liver inflammation by regulating the gene expression of *MARS1*, *CDH11* and their related signal pathways, thereby preventing the occurrence of liver diseases.

The GO pathway enrichment analysis of the differential genes between RPC and RPM showed that the enriched pathways were mainly associated with the metabolic processes of lipid, cytokine, carboxylic acid, organic acid, oxoacid, and the activity of fatty acid omega-hydroxylase, steroid hydroxylase, oxidoreductase, chemokine, etc. The different metabolic pathways enriched by the KEGG pathway indicated that the differences between choline and nicotinamide in the regulation of liver metabolism were mainly manifested in fat metabolism, biosynthesis, drug metabolism, etc. This may be due to the fact that choline can also regulate the metabolism process of fat and energy in the liver as well as related signals of oxidative stress and inflammation as a signal, in addition to catabolizing into other substances and then participating in membrane biosynthesis, fat metabolism and transportation, and other substance synthesis [14,75]. Nicotinamide is mostly used to synthesize NAD for energy supply after entering the liver, to maintain the level of reduced glutathione and thioredoxin in the antioxidant system, and to participate in cell energy metabolism and other processes.

## 5. Conclusions

In this study, the effects of RPC and RPM on the liver transcriptome of perinatal cows were investigated, and 1123 DEGs and 27 different KEGG pathways were identified, mainly including retinol metabolism, metabolism of xenobiotics by cytochrome P450, steroid hormone biosynthesis, pyrimidine metabolism, bile secretion, linoleic acid metabolism, arachidonic acid metabolism, chemokine signaling pathway, aminoacyl-tRNA biosynthesis, purine metabolism, glutathione metabolism, etc. The expression levels of genes related to liver metabolism were identified, and it was found that, compared with the niacinamide group, the choline group could up-regulate the expression levels of *FGF21*, *CYP26A1*, *SLC13A5*, *SLCO1B3*, *FBP2*, *MARS1*, and *CDH11*. Overall, this study shows that RPC plays a prominent role in liver metabolism by regulating metabolic processes such as fatty acid synthesis and metabolism and glucose metabolism. Yet, RPM is more involved in biological processes such as the TCA cycle, ATP generation and inflammatory signaling.

## Figures and Tables

**Figure 1 metabolites-13-00594-f001:**
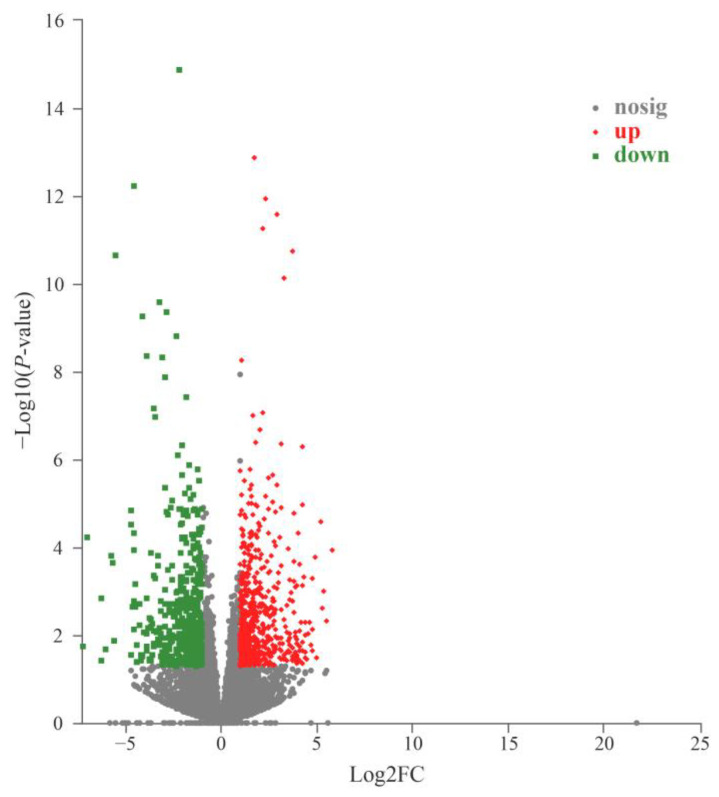
Volcano plot of DEGs. The corrected *p*-value is *p*−adjust. Each dot represents a gene in which red represents significant up-regulation of gene expression and green represents significant down-regulation of gene expression. The closer the point is to the two sides and the upper side, the more significant the difference is.

**Figure 2 metabolites-13-00594-f002:**
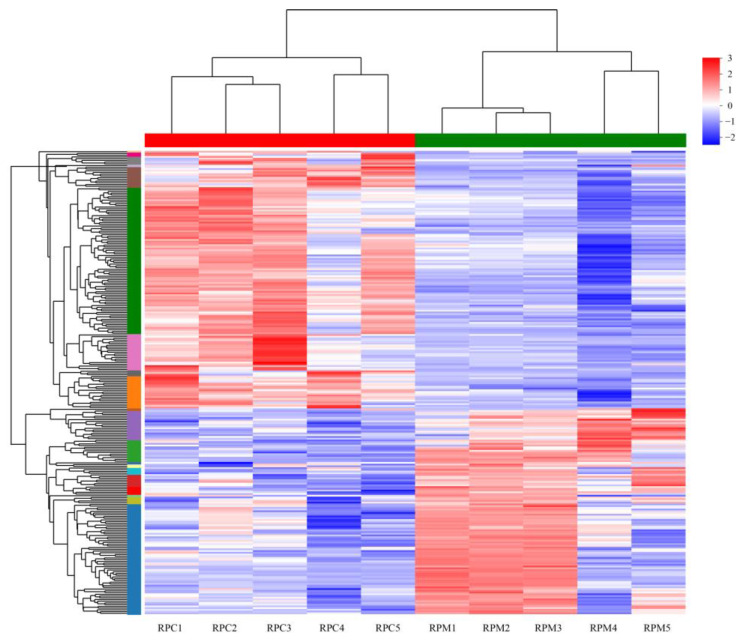
Cluster heatmap of DEGs in different samples. Different colors represent the expression value of the gene after standardization in each sample. On the left side is the tree diagram of the gene cluster and the module diagram of the sub cluster. The closer the two gene branches are, the closer their expression amounts are. The upper side in this graph is the tree diagram of the sample cluster. The closer the two samples branch, the closer the expression patterns of all genes in the two samples are. RPC = rumen-protected choline group; RPM = rumen-protected nicotinamide group.

**Figure 3 metabolites-13-00594-f003:**
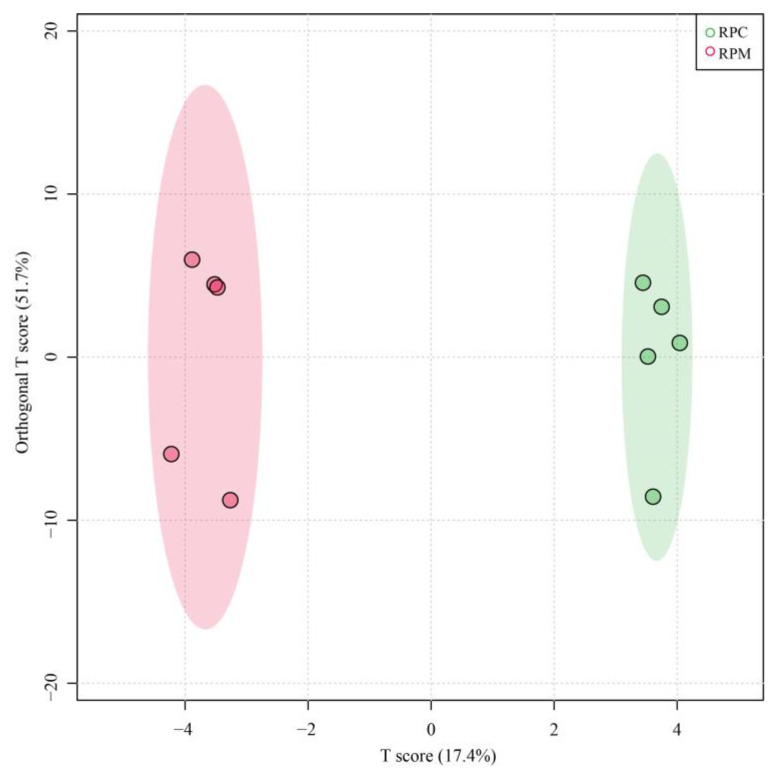
OPLS-DA score plot of different samples. RPC = rumen-protected choline group; RPM = rumen-protected nicotinamide group. RPC = rumen-protected choline group; RPM = rumen-protected nicotinamide group.

**Figure 4 metabolites-13-00594-f004:**
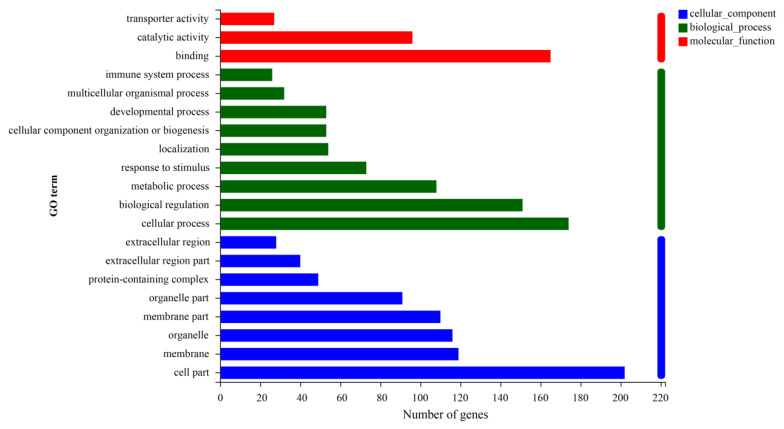
Top 20 GO terms in gene abundance. The ordinate in the graph represents the GO’s secondary classification term, the lower abscissa represents the number of genes/transcripts for that secondary classification on the alignment. BP = biological process; CC = cellular component; MF = molecular function.

**Figure 5 metabolites-13-00594-f005:**
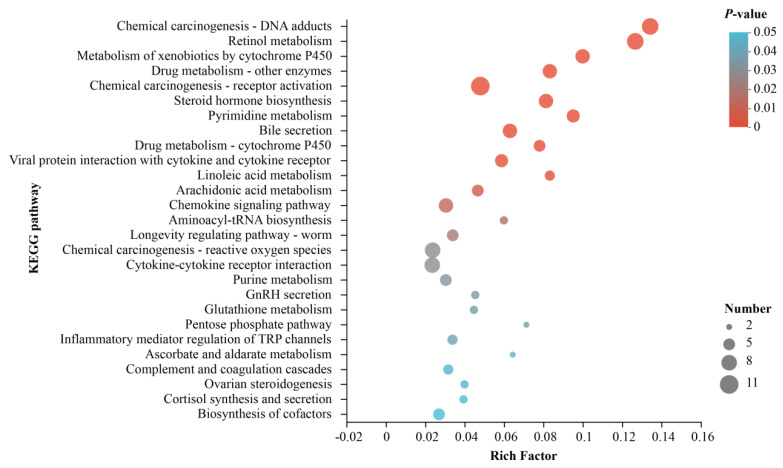
Enriched KEGG pathway of DEGs. The rich factor is the ratio of differentially methylated and expressed gene numbers annotated in this pathway term to all gene numbers annotated in this pathway term. The size of the dot represents the number of genes in the path, and the color of the dot corresponds to different *p*-value ranges.

**Figure 6 metabolites-13-00594-f006:**
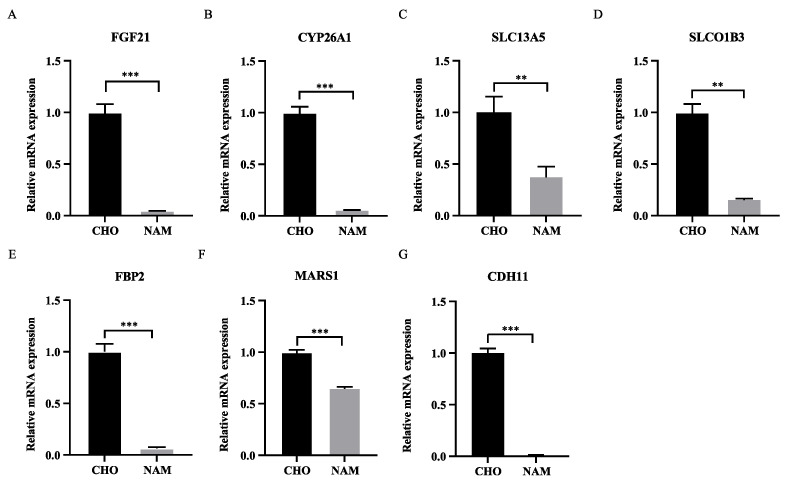
Verification of gene expression. (**A**) mRNA expression of *FGF21*. (**B**) mRNA expression of *CYP26A1*. (**C**) mRNA expression of *SLA13A5*. (**D**) mRNA expression of *SLCO1B3*. (**E**) mRNA expression of *FBP2*. (**F**) mRNA expression of *MARS1*. (**G**) mRNA expression of *CDH11*. ** indicates *p* < 0.01, *** indicates *p* < 0.001.

**Table 1 metabolites-13-00594-t001:** Sequences of primers.

Target Gene	Forward (5′-3′)	Reverse (5′-3′)
*FBP2*	GTCACGTTAACGCTTCCTGC	CTGCACTGCTGGCGTTTTAG
*SLCO1B3*	CACACTTGGGTGAATGCCCA	ATGTGGTACCTCCTGTTGCAG
*MARS1*	GGGCTTCCAGCTGATGCTAT	TGGACTCTCTGTAGCCACCA
*FGF21*	ATCGCTCCACTTTGACCCTG	GGGCTTCGGACTGGTAAACA
*CYP26A1*	CCCTATGCTGTGGCTGCAAT	CCAAGGGCTGACAAACTGGA
*CDH11*	GGGCTTCCAGCTGATGCTAT	TGGACTCTCTGTAGCCACCA
*SLC13A5*	TGATGACGTAGGCACACCTG	TTGACAATCCATGCCAGCCG
*β-actin*	GCACTCTTCCAGCCTTCCTT	AATGCCAGGGTACATGGTGG

**Table 2 metabolites-13-00594-t002:** The reaction system and condition of RT-PCR.

Reagent Name	Volume (μL)
2× SYBR^®^ Green Pro Taq HS Premix	5.0
Forward primer (10 μM)	0.5
Reverse primer (10 μM)	0.5
cDNA	4.0
Reaction conditions: 95 °C 30 s; 2 95 °C 5 s, 60 °C 30 s (40 cycles) (two-step algorithm) Dissociation stage: 95 °C 10 s, 65 °C 60 s, 97 °C 1 s	

**Table 3 metabolites-13-00594-t003:** Top 30 DEGs identified in RPC and RPM groups.

Gene Name	Gene Description	Log_2_FC (RPC/RPM) *	*p*-Value
*FARS2*	phenylalanyl-tRNA synthetase 2, mitochondrial	1.757	1.31 × 10^13^
*FGF21*	fibroblast growth factor 21	−4.535	5.77 × 10^13^
*NOTUM*	notum, palmitoleoyl-protein carboxylesterase	2.310	1.13 × 10^12^
*CYP2C19*	cytochrome P450, family 2, subfamily C, polypeptide 19	2.894	2.58 × 10^12^
*KCNN2*	potassium calcium-activated channel subfamily N member 2	2.163	5.49 × 10^12^
*FUT1*	fucosyltransferase 1	−5.524	2.32 × 10^12^
*CYP26A1*	cytochrome P450, family 26, subfamily A, polypeptide 1	3.315	7.46 × 10^11^
*CDH11*	cadherin 11	−3.201	2.64 × 10^10^
*DTX1*	deltex E3 ubiquitin ligase 1	−2.872	4.29 × 10^10^
*ATP6V1C2*	ATPase H+ transporting V1 subunit C2	−4.079	5.57 × 10^10^
*SLC13A5*	solute carrier family 13 member 5	−2.369	1.53 × 10^9^
*LRRC73*	leucine-rich repeat-containing 73	−3.907	4.41 × 10^9^
*HOPX*	HOP homeobox	−3.070	4.68 × 10^9^
*SLCO1B3*	solute carrier organic anion transporter family member 1B3	1.089	5.55 × 10^9^
*MICAL2*	microtubule-associated monooxygenase, calponin and LIM domain-containing 2	−2.959	1.36 × 10^8^
*GPC3*	glypican 3	-1.830	3.69 × 10^8^
*ADCY2*	adenylate cyclase 2	−3.537	7.04 × 10^7^
*HRG*	histidine-rich glycoprotein	1.666	1.02 × 10^7^
*GLI1*	GLI family zinc finger 1	−3.446	1.10 × 10^7^
*CES1*	carboxylesterase 1 (monocyte/macrophage serine esterase 1)	2.000	2.08 × 10^7^
*CD1D*	CD1D antigen, d polypeptide	1.810	4.24 × 10^7^
*FBP2*	fructose-bisphosphatase 2	3.167	4.36 × 10^7^
*MARS1*	methionyl-tRNA synthetase 1	−2.070	4.71 × 10^7^
*SEC14L3*	SEC14-like lipid binding 3	4.21	5.31 × 10^7^
*INHBE*	inhibin subunit beta E	−2.277	8.08 × 10^7^
*CARS1*	cysteinyl-tRNA synthetase 1	−1.218	1.71 × 10^6^
*PLEK*	pleckstrin	1.545	1.72 × 10^6^
*TKT*	transketolase	1.009	1.75 × 10^6^
*CITED4*	Cbp/p300-interacting transactivator with Glu/Asp-rich carboxy-terminal domain 4	−2.026	2.22 × 10^6^
*BIRC5*	baculoviral IAP-repeat-containing 5	2.703	2.34 × 10^6^

* RPC = rumen-protected choline; RPM = rumen-protected nicotinamide.

**Table 4 metabolites-13-00594-t004:** Top 20 enriched GO terms among DEGs. BP = biological process; CC = cellular component; MF = molecular function.

GO ID	Category	Description	*p*-Value	Counts
GO:0006629	BP	lipid metabolic process	1.18 × 10^6^	29
GO:0044281	BP	small molecule metabolic process	1.56 × 10^6^	40
GO:0034754	BP	cellular hormone metabolic process	3.39 × 10^6^	8
GO:0006805	BP	xenobiotic metabolic process	4.33 × 10^6^	7
GO:0005615	CC	extracellular space	2.60 × 10^6^	33
GO:0004497	MF	monooxygenase activity	3.61 × 10^6^	10
GO:0005737	CC	cytoplasm	5.73 × 10^6^	88
GO:0006082	BP	organic acid metabolic process	7.52 × 10^6^	23
GO:0043436	BP	oxoacid metabolic process	1.41 × 10^5^	22
GO:0042445	BP	hormone metabolic process	1.64 × 10^5^	9
GO:0019752	BP	carboxylic acid metabolic process	1.78 × 10^5^	21
GO:0110165	CC	cellular anatomical entity	2.00 × 10^5^	236
GO:0032787	BP	monocarboxylic acid metabolic process	3.31 × 10^5^	15
GO:0016712	MF	oxidoreductase activity, acting on paired donors, with incorporation or reduction in molecular oxygen, reduced flavin or flavoprotein as one donor, and incorporation of one atom of oxygen	2.96 × 10^5^	5
GO:0046906	MF	tetrapyrrole binding	3.25 × 10^5^	10
GO:0008009	MF	chemokine activity	4.86 × 10^5^	5
GO:0008202	BP	steroid metabolic process	8.63 × 10^5^	9
GO:0098754	BP	detoxification	9.17 × 10^5^	7
GO:0102033	MF	fatty acid omega-hydroxylase activity	9.08 × 10^5^	2
GO:0008395	MF	steroid hydroxylase activity	9.33 × 10^5^	5

**Table 5 metabolites-13-00594-t005:** Top 20 enriched KEGG pathways of DEGs.

Pathway ID	KEGG Pathways	*p*-Value	Counts
map00830	Retinol metabolism	5.95 × 10^8^	9
map05204	Chemical carcinogenesis - DNA adducts	3.54 × 10^8^	9
map00980	Metabolism of xenobiotics by cytochrome P450	9.19 × 10^6^	7
map00140	Steroid hormone biosynthesis	3.57 × 10^5^	7
map05207	Chemical carcinogenesis - receptor activation	3.44 × 10^5^	11
map00983	Drug metabolism - other enzymes	3.07 × 10^5^	7
map00240	Pyrimidine metabolism	5.39 × 10^5^	6
map04976	Bile secretion	0.00018	7
map00982	Drug metabolism - cytochrome P450	0.00058	5
map04061	Viral protein interaction with cytokine and cytokine receptor	0.00076	6
map00591	Linoleic acid metabolism	0.00165	4
map00590	Arachidonic acid metabolism	0.00563	5
map04062	Chemokine signaling pathway	0.01122	7
map00970	Aminoacyl-tRNA biosynthesis	0.01591	3
map04212	Longevity regulating pathway - worm	0.02022	5
map04060	Cytokine-cytokine receptor interaction	0.02835	8
map04750	Inflammatory mediator regulation of TRP channels	0.03684	4
map00030	Pentose phosphate pathway	0.03520	2
map05208	Chemical carcinogenesis - reactive oxygen species	0.02709	8
map00230	Purine metabolism	0.03055	5

## Data Availability

The data presented in this study are available in the article and the Appendix A.

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
