# Peer review of "The Effects of Rumen-Protected Choline and Rumen-Protected Nicotinamide on Liver Transcriptomics in Periparturient Dairy Cows"

_metabolites, 2023, doi:10.3390/metabo13050594_

Round 1

Reviewer 1 Report (Previous Reviewer 1)

The author has carefully revised the manuscript based on the reviewer's comments.I suggest checking the manuscript again. In Citation, there may be spelling errors in the name, such as Chuanjaing Cai*.

Author Response

Thank you for your valuable modification comments. Based on the modification comments, we have made the following modifications.

Point 1: The author has carefully revised the manuscript based on the reviewer's comments.I suggest checking the manuscript again. In Citation, there may be spelling errors in the name, such as Chuanjaing Cai*.

Response: We have reviewed the entire text and checked and modified the spelling of the author's name.

Reviewer 2 Report (Previous Reviewer 2)

General Comments

This study improve the knowledge on the field. I think that the subject of the work is of interest and that the topic of the manuscript is appropriate for the Journal. The information is of significant interest to the Journal's readers. However I suggest some correction throughout the manuscript. In particular, I think that introduction could be improved by better stressing the features of peripartum period in mammals. Moreover, several missing information should be added in the methods section.

Specific Comments

The title well reflects the main aim and findings of the work.

I suggest avoiding use of personal form (i.e. our, we, etc.) throughout the manuscript.

The abstract adequately summarize results and significance of the study. However, methodology description need to be improved. Authors should indicate the statistical analysis applied on the obtained data and related p values in results description.  

Keywords represent the article adequately.

The introduction section is well written and it falls within the topic of the study, and Authors cited appropriately bibliographic information. I think that Authors should better emphasize the features of peripartum period in mammals with focus on physiological adaptation of animals during this particular life phase. On this regards, I suggest to enhance the first sentence of introduction “Perinatal cows (21 days respectively before and after parturition) are subject to the imbalance between energy demand and dietary energy intake, resulting in stimulation of fat mobilization [1] and an elevated level of non-esterified fatty acids (NEFA) in the body.” As following “The peripartum period, lasting from 3 weeks before and 3 weeks after parturition, induces remarkable physiological and metabolic adaptations in mammals which are crucial for a good reproductive performance and to ensure the suitable development of the foetus and to provide adequate substrates that are needed in utero and following birth (Bazzano M et al., Reproduction in domestic animals, 2014, 49(6) 947–953; Arfuso F. et al., Fiore E. et al., Archiv Tierzucht, 2014, 57:1-9). During peripartum period, dairy cows face a dysfunctional immune system and an increased inflammatory state. A modulation of pathways related to metabolism, immune status and endocrine system (Arfuso F. et al., Theriogenology 196, 2023, 157-166).  Periparturient cows are subject to the imbalance between energy demand and dietary energy intake, resulting in stimulation of fat mobilization [1] and an elevated level of non-esterified fatty acids (NEFA) in the body.”

The section of Materials and Methods is clear for the reader and it well describes the methods applied in the study. However, some information should be added.

What about the health status of animals during the monitoring period? What about the calving? Did cows deliver healthy, viable full-term calves, without assistance? What about age, of enrolled cows? Please add more information about this aspect of methodology.

Lines 94-95, Authors wrote “Liver biopsy was taken on the 21st day postmortem” I think it was a mistake, postmortem??? Should it postpartum??? Please check and correct.

Regarding statistical analysis, Did Authors perform a normality test in order to test the normal distribution of data? Please clarify this aspect.

Results section as well as Discussion section is clear and well written, the findings obtained in the study were well discussed and justified with appropriate references.

The conclusion section is too short resulting unclear. Authors should rewrite it summarizing the topic of the study, better describing (briefly) the results and better emphasizing the significance of the study.

Tables and figures are generally good and well represent the results of the study.

Authors should check and standardize the references in the list according to journal guidelines.

Author Response

Thank you for your valuable modification comments. Based on the modification comments, we have made the following modifications.

  1. I suggest avoiding use of personal form (i.e. our, we, etc.) throughout the manuscript.

Response: We reviewed the manuscript and revised the using of personal form.

  1. The abstract adequately summarize results and significance of the study. However, methodology description need to be improved. Authors should indicate the statistical analysis applied on the obtained data and related p values in results description.

Response: Line 22. We supplemented the analysis of gene expression level and related p values in the abstract.

  1. The introduction section is well written and it falls within the topic of the study, and Authors cited appropriately bibliographic information. I think that Authors should better emphasize the features of peripartum period in mammals with focus on physiological adaptation of animals during this particular life phase. On this regards, I suggest to enhance the first sentence of introduction “Perinatal cows (21 days respectively before and after parturition) are subject to the imbalance between energy demand and dietary energy intake, resulting in stimulation of fat mobilization [1] and an elevated level of non-esterified fatty acids (NEFA) in the body.” As following “The peripartum period, lasting from 3 weeks before and 3 weeks after parturition, induces remarkable physiological and metabolic adaptations in mammals which are crucial for a good reproductive performance and to ensure the suitable development of the foetus and to provide adequate substrates that are needed in utero and following birth (Bazzano M et al., Reproduction in domestic animals, 2014, 49(6) 947–953; Arfuso F. et al., Fiore E. et al., Archiv Tierzucht, 2014, 57:1-9). During peripartum period, dairy cows face a dysfunctional immune system and an increased inflammatory state. A modulation of pathways related to metabolism, immune status and endocrine system (Arfuso F. et al., Theriogenology 196, 2023, 157-166). Periparturient cows are subject to the imbalance between energy demand and dietary energy intake, resulting in stimulation of fat mobilization [1] and an elevated level of non-esterified fatty acids (NEFA) in the body.

Response: Line 32. According to the comments, the background introduction and references of perinatal cows in the introduction have been supplemented.

  1. The section of Materials and Methods is clear for the reader and it well describes the methods applied in the study. However, some information should be added. What about the health status of animals during the monitoring period? What about the calving? Did cows deliver healthy, viable full-term calves, without assistance? What about age, of enrolled cows? Please add more information about this aspect of methodology.

Response: During the monitoring period, the cows in this experiment, in health status and 1-2 parity, delivered healthy, viable full-term calves. The relevant contents have been supplemented in point 2.1 ( Animals and Sample Acquisition).

  1. Lines 94-95, Authors wrote “Liver biopsy was taken on the 21st day postmortem” I think it was a mistake, postmortem??? Should it postpartum??? Please check and correct.

Response: Line 82. Our study used liver biopsy technology to collect liver samples from perinatal cows. In new manuscript, we revised the method of liver sample collection in point 2.1 (animals and sample acquisition) and supplemented the statement about animal welfare and authorization.

  1. Regarding statistical analysis, Did Authors perform a normality test in order to test the normal distribution of data? Please clarify this aspect.

Response: Line 159. We performed a normality to test the normal distribution of data. The relevant contents have been supplemented in point 2.5 (Statistical Analysis).

  1. The conclusion section is too short resulting unclear. Authors should rewrite it summarizing the topic of the study, better describing (briefly) the results and better emphasizing the significance of the study.

Response: Rewritten the conclusion section to more accurately describe the research results.

  1. Authors should check and standardize the references in the list according to journal guidelines.

Response: Revised references list according to journal guidelines.

Reviewer 3 Report (Previous Reviewer 3)

Also if the authors made a deep revision process of their manuscript and they add Real Time PCR determinations to validate some oftheir RNAseq findings, there are still many phrases that are not clear and sometimes, revision and rephrasing process made by the author does not help to clarify.

Here an example taken from point 2.1 (animals and sample acquisition), lines 94-95:

Liver biopsy was taken on the 21st day postmortem (collected at 13:00 each time), snap frozen and stored at - 80.

The original phrase taken from the first version of the manuscript was:

Liver biopsy was performed on the 21st day after delivery (collected at 13:00 each time), and liver tissue samples were collected by biopsy needle puncture. The liver samples collected were quickly stored at - 80. All the experimental procedures in the present study were conducted according to the Animal Protection Law based on the Guide for the Care and Use of Laboratory Animals approved by the Ethics Committee of Northwest A&F University.

The statement about animal welfare and authorization disappeared in the revised version of the manuscript and, moreover, it seems now that animals were euthanized, being reported that specimens were collected postmortem. If sample collection was made by biopsy techniques (as further reported in supplementary materials) how it was possible to made postmortem samplings??

I strongly recommend a complete proofreading of the manuscript by a native English reviewer before to consider the publication of this manuscript. There are still too many sentences that are not clear and straightforward.

Here followed other comments to be addressed by the authors:

- line 114: authors report the use of ViiA 7 Real-Time PCR System for their PCR analysis, but then in brackets is reported the wrong name of supplier together with another Thermal cycler instrument (Roche LightCycler® 480). Please correct and specify if the ViiA 7 or the Lightcycler were used for such PCR determinations.

- line 115: the authors choose Beta-actin as reference gene for normalization of their gene expression study. How was the test and selection of reference genes made? How did they choose this target? Please add reference or report some data about its stability (for example genorm and/or normfinder tests).

- line 116: in table 1 together with primer sequences please add melting temperatures being Sybr green detection methods used.

- line 167: volcanic plot term is not correct. Please change it with Volcano Plot

- Figure 2: In liver heatmap plot the clustering was made using 5 samples for each group (RCP and RPM respectively). In animal trial data section 6 animal for each group are reported. Please explain why one animal for each group is missing.

- It is not clear why the authors choose to validate some DEG of their RNAseq experiments in human liver cells (LO2) using CHO and NAM treatments. A validation by Real Time PCR performed on the same liver biopsies considering top up- and down-regulated DEG should have been sufficient. Did the authors no longer have the RNA extracts from the liver biopsies available for further analysis? Please add comments and considerations regarding the choose of an in vitro experiment on human liver cell lines.

Author Response

Thank you for your valuable modification comments. Based on the modification comments, we have made the following modifications.

1.lines 94-95:

Liver biopsy was taken on the 21st day postmortem (collected at 13:00 each time), snap frozen and stored at - 80.

The original phrase taken from the first version of the manuscript was:

Liver biopsy was performed on the 21st day after delivery (collected at 13:00 each time), and liver tissue samples were collected by biopsy needle puncture. The liver samples collected were quickly stored at - 80℃. All the experimental procedures in the present study were conducted according to the Animal Protection Law based on the Guide for the Care and Use of Laboratory Animals approved by the Ethics Committee of Northwest A&F University.

Response: Line 82. Our study used liver biopsy technology to collect liver samples from perinatal cows. In new manuscript, we revised the method of liver sample collection in point 2.1 (animals and sample acquisition) and supplemented the statement about animal welfare and authorization.

2.line 114:

authors report the use of ViiA 7 Real-Time PCR System for their PCR analysis, but then in brackets is reported the wrong name of supplier together with another Thermal cycler instrument (Roche LightCycler® 480). Please correct and specify if the ViiA 7 or the Lightcycler were used for such PCR determinations.

Response: Line 99. Our study used Roche LightCycler® 480 Real-Time PCR System for PCR determinations in point 2.2 (Culture of Human Hepatocyte Line LO2), and the manuscript has been revised.

3.line 115:

the authors choose Beta-actin as reference gene for normalization of their gene expression study. How was the test and selection of reference genes made? How did they choose this target? Please add reference or report some data about its stability (for example genorm and/or normfinder tests).

Response: This study referred to literature from others and selected Beta-actin, as an internal reference gene, has been supplemented in the literature.

4.line 116:

in table 1 together with primer sequences please add melting temperatures being Sybr green detection methods used.

Response: Line 120. We supplemented the PCR reaction system and methods (Table 2) in point 2.3 (RNA isolation and quantitative real-time polymerase chain reaction).

5.line 167:

volcanic plot term is not correct. Please change it with Volcano Plot

Response: Line 169. Modify the “Volcanic” to the “Volcano”.

6.Figure 2:

In liver heatmap plot the clustering was made using 5 samples for each group (RCP and RPM respectively). In animal trial data section 6 animal for each group are reported. Please explain why one animal for each group is missing.

Response: Each treatment group in this experiment used 5 experimental animals, which have been revised in the manuscript.

7.

It is not clear why the authors choose to validate some DEG of their RNAseq experiments in human liver cells (LO2) using CHO and NAM treatments. A validation by Real Time PCR performed on the same liver biopsies considering top up- and down-regulated DEG should have been sufficient. Did the authors no longer have the RNA extracts from the liver biopsies available for further analysis? Please add comments and considerations regarding the choose of an in vitro experiment on human liver cell lines.

Response: We have conducted extensive experiments to verify the fat deposition status of LO2 liver cells after the addition of NEFA. The measurement of triglycerides, non esterified fatty acids, β- hydroxybutyric acid, cell survival rate, lactate dehydrogenase and other indicators has shown that LO2 cells with NEFA addition can serve as fat deposition model cells. Due to other reasons, the sequencing company did not return the remaining RNA sample, so we used LO2 cells for gene expression verification. We have supplemented the reasons for the selection of LO2 cells in the manuscript.

Round 2

Reviewer 3 Report (Previous Reviewer 3)

None

This manuscript is a resubmission of an earlier submission. The following is a list of the peer review reports and author responses from that submission.

Round 1

Reviewer 1 Report

This is a well-designed study and it was carried out according to standard protocols. The manuscript is generally well written with some minor grammar error. In this study, the liver samples collected by liver puncture biopsy were used to investigate the changes of liver metabolism in perinatal dairy cows with the addition of rumen protected choline and rumen protected nicotinamide. Through transcriptomic analysis, it was found that there were significant differences in lipid metabolism and oxidative stress between the two groups. The differences of choline and nicotinamide in liver metabolism were described, which laid a foundation for the future research on the functions and mechanisms of these two nutrients. The respective requirements, and recommendations are as follows:

1.Line 30-101: Consider whether the part of "Introduction" is too long.

2.Line 66, 86: Delete the space between the inserted document and the text.

3.Line 113: "total mixed ratio" to total mixed ration.

5.Line 116: Delete the space between "-" and "80℃".

6.Line 147: Please mention any special command/flag added in the bioinformatics analysis tools. Please mention the statement as "all default parameters were used unless otherwise mentioned".

7.Line 157, 164, 167: "Table S#" to "Supplementary Table S#"; "Figure S#" to "Supplementary Figure S#".

8.Line 248: It is suggested that only the top ten KEGG enrichment pathways should be shown in the detail table.

Reviewer 2 Report

It is well known that the periparturient period is a metabolically challenging period for dairy cows because homeorhetic changes, negative energy balance, and increased risk for metabolic disorders can negatively affect dairy cattle health and productivity. Many studies are currently available in scientific literature on the practice adopted to prevent hepatic lipidosis, control BCS loss (Janovick et al., 2011) across the transition to lactation period, feed prepartum diets with decreased dietary energy (DE) density, and supplement nutrients that promote liver function such as choline (Overton and Piepenbrink, 1999; Arshad et al., 2020; McFadden et al., 2020). Nutrients such as rumen-protected choline (RPC) and/or nicotinamide are supplemented during the transition period, and research suggests that RPC supplementation decreases FA accumulation in the liver in a dose-responsive manner (Zenobi et al., 2018) and increases milk and ECM production (Arshad et al., 2020; McFadden et al., 2020; Caprarulo et al., 2020; Tienken et al., 2015). Authors wrote that the difference between choline and nicotinamide on liver metabolism and its mechanism in perinatal dairy cows is still unclear, However, I think that they enrolled a low number of animals to test their hypothesis.

Reviewer 3 Report

The whole paper should be carefully revised in its English form. The structure of phrases is often not clear and straightforward as it needs to be in a scientific manuscript. There is a lot of incorrect terminology, and several terms seem used not properly (for example in line 121, 123, 125 the term “database” should be referred to “libraries” to be sequenced, the term “volcanic map” in line 140 and 165 should be better replaced by “Volcano plot”, etc..).

Moreover, from a scientific point of view the list of DEG found by authors need to be verified at least on more significant up- and/or down-regulated genes by Real Time PCR, in order to confirm what recorded by RNAseq analysis.

I cannot recommend the publication of present manuscript in its current form. An extensive revision of English language and at least a confirmatory analysis on DEG must be performed before to consider and discuss the results described by the authors.